# Split-Attention U-Net: A Fully Convolutional Network for Robust Multi-Label Segmentation from Brain MRI

**DOI:** 10.3390/brainsci10120974

**Published:** 2020-12-11

**Authors:** Minho Lee, JeeYoung Kim, Regina EY Kim, Hyun Gi Kim, Se Won Oh, Min Kyoung Lee, Sheng-Min Wang, Nak-Young Kim, Dong Woo Kang, ZunHyan Rieu, Jung Hyun Yong, Donghyeon Kim, Hyun Kook Lim

**Affiliations:** 1Research Institute, NEUROPHET Inc., Seoul 06247, Korea; minho.lee@neurophet.com (M.L.); reginaeunyoungkim@neurophet.com (R.E.K.); clarence@neurophet.com (Z.R.); yong@neurophet.com (J.H.Y.); 2Department of Radiology, Eunpyeong St. Mary’s Hospital, College of Medicine, The Catholic University of Korea, Seoul 03312, Korea; jeeyoungkim@catholic.ac.kr (J.K.); catharina@catholic.ac.kr (H.G.K.); oasis1979@gmail.com (S.W.O.); 3Institute of Human Genomic Study, College of Medicine, Korea University, Ansan 15355, Korea; 4Department of Psychiatry, University of Iowa, Iowa City, IA 52242, USA; 5Department of Radiology, Yeouido St. Mary’s Hospital, College of Medicine, The Catholic University of Korea, Seoul 07345, Korea; 22000659@cmcnu.or.kr; 6Department of Psychiatry, Yeouido St. Mary’s Hospital, College of Medicine, The Catholic University of Korea, Seoul 07345, Korea; smwang11@naver.com (S.-M.W.); nakyoung17@gmail.com (N.-Y.K.); 7Department of Psychiatry, Seoul St. Mary’s Hospital, College of Medicine, The Catholic University of Korea, Seoul 06591, Korea; kato7@hanmail.net

**Keywords:** multi-label brain segmentation, split-attention block, deep learning, fine-tuning, SAU-Net

## Abstract

Multi-label brain segmentation from brain magnetic resonance imaging (MRI) provides valuable structural information for most neurological analyses. Due to the complexity of the brain segmentation algorithm, it could delay the delivery of neuroimaging findings. Therefore, we introduce Split-Attention U-Net (SAU-Net), a convolutional neural network with skip pathways and a split-attention module that segments brain MRI scans. The proposed architecture employs split-attention blocks, skip pathways with pyramid levels, and evolving normalization layers. For efficient training, we performed pre-training and fine-tuning with the original and manually modified FreeSurfer labels, respectively. This learning strategy enables involvement of heterogeneous neuroimaging data in the training without the need for many manual annotations. Using nine evaluation datasets, we demonstrated that SAU-Net achieved better segmentation accuracy with better reliability that surpasses those of state-of-the-art methods. We believe that SAU-Net has excellent potential due to its robustness to neuroanatomical variability that would enable almost instantaneous access to accurate neuroimaging biomarkers and its swift processing runtime compared to other methods investigated.

## 1. Introduction

Magnetic resonance imaging (MRI) provides detailed in vivo insights into the human brain’s morphology, which are essential for development, aging, and disease research [1,2,3,4]. To access measurements such as volume, thickness, or shape for structures, neuroanatomy must be segmented, which is a time-consuming issue when it is manually annotated. A manual delineation of neuroanatomy on MRI scans has long been regarded as the “ground truth”. Therefore, fully automation-based algorithms have been sought to reduce the manual effort. In the past, many whole-brain segmentation methods were proposed, including, but not restricted, to the region growing, clustering, and template-based deformation methods [5]. Template-based segmentation is an essential segmentation method that applies tissue labels to the unlabeled images using structural brain MRI as well as the corresponding manual segmentation. In the template-based segmentation models, the deformable registration methods typically operate by spatially deforming a template to fit a previously unlabeled target image [6,7]. A single template-based segmentation has been successfully applied in some commercial applications [6,7]. However, segmentation with a single template performs poorly when the target image has considerable inter-patient variation in neuroanatomy [8,9]. More studies have proposed multi-atlas-based schemes, which have become the actual standard template-based segmentation approach [10,11]. The multi-atlas-based segmentation methods can segment the brain from a clinically obtained T1-weighted MRI volume into more than 100 structural labels using a small number of manually annotated and representative scans. In practice, the multi-atlas segmentation methods are regarded as the standard for whole-brain segmentation methods because of their excellent accuracy and reproducibility. The patch-based methods have been proposed to handle the local variations from incomplete registrations [12,13,14,15,16]. Consequently, the multi-atlas label fusion-based methods have been studied to model the spatial relationships between atlases and target images in the three-dimension patches. However, one of the critical limitations of the conventional multi-atlas segmentation methods is their high complexity cost. Therefore, a multi-atlas label fusion is typically performed with a small number of templates. Many previous studies have been proposed to develop faster and robust multi-atlas segmentation approaches using unlabeled or automatically labeled data to utilize a larger number of templates or even previously unlabeled MRI images [17,18]. The machine learning-based methods have been incorporated into the multi-atlas segmentation method to replace fusion based on the traditional voting or statistical patch approaches. Several machine learning-based multi-atlas label fusion approaches have been developed to learn multi-atlas segmentation from a larger number of atlas templates [19,20,21,22,23]. The core idea of creating a deep-learning architecture from training image patches has contributed to and inspired various convolutional neural network (CNN) methods, including our proposed method.

Recently, deep-learning methods have been widely developed for whole-brain segmentation with multiple labels. The most straightforward strategy for segmenting the whole brain is to input the entire brain volume to a three-dimensional CNN-based segmentation framework, like U-Net [24] or V-Net [25]. However, it is inefficient to apply clinically used MRI images in State-Of-The-Art (SOTA) three-dimensional fully convolutional networks (FCNs) because the amount of memory in standard graphics processing units (GPUs) cannot handle the high resolution of the MRI images. Another challenge of utilizing CNN methods is that manually traced whole-brain MRI images with detailed structural annotations are rare commodities. Many studies have been developed to challenge the limits placed on training data by GPU memory restrictions. De Brebisson’s method proposed an integrated CNN method to learn two three-dimensional patches and their spatial coordinates for multi-label whole-brain segmentation [26]. This approach was subsequently extended to BrainSegNet [27], which employed 2.5-dimensional patches for training an FCN. Recently, DeepNAT [28] was proposed to perform multi-task learning on the three-dimensional patches. These methods design the whole brain segmentation as a per-voxel segmentation problem. More recently, more robust FCNs have been introduced to multi-label whole-brain segmentation from another perspective. QuickNAT is a two-dimensional approach developed to train an FCN framework using a number of auxiliary labels on initially unlabeled data [29]. Although the training was in a two-dimensional manner, this approach revealed a promising direction for leveraging the whole-brain segmentation architecture, using manually traced images and initially unlabeled data. FastSurfer developed a module containing competitive dense blocks and skip pathways for training multi-slice-based 2D CNNs [30].

However, the two-dimension-based segmentation approaches typically return low spatial consistency in the third dimension. Therefore, it would be fascinating to perform a three-dimensional FCN for whole-brain segmentation for higher spatial consistency [24,25]. Recently, multi-task learning [31] and semi-supervised learning [32] have been performed for whole-brain segmentation. Among such work, Rajchl et al. proposed a novel three-dimensional multi-task learning network with remarkably decent segmentation performance [31]. However, the network’s input image size is restricted to 128 × 128 × 128 because of the GPU memory restrictions, which misses about two-thirds of the spatial information available in a 1 mm isotropic brain volume in the Montreal Neurological Institute (MNI) template space [33]. Therefore, directly applying the three-dimensional FCN to whole-brain segmentation is still limited by current GPU capabilities. To overcome such challenges, Li et al. proposed a 3D CNN-based sliding window-based method, which utilized a single network to learn the patches at different spatial locations [34]. Additionally, the spatially localized atlas network tiles (SLANT) method was proposed to improve patch-based training using multiple independent 3D FCN methods [35].

Here, we propose Split-Attention U-Net (SAU-Net), a deep-learning architecture that can segment the whole brain into 33 structural labels in just 20–30 s on the GPU. The basic architecture is inspired by ResNeSt [36] and U-Net++ [37]. By applying the split-attention module of ResNeSt to the output of each pyramid level of the U-Net++ structure and evolving normalization instead of batch normalization (BN) [38] and rectified linear unit (ReLU) [39], the class differentiation between feature maps of the level is improved, and efficient three-dimensional training with a small number of batch sizes is possible. It is possible to train it more efficiently and it has strengths in the inference speed. Moreover, it can be known that the three-dimensional patch-based SAU-Net has a higher level of spatial information for accurate segmentation of neuroanatomical structures when compared to QuickNAT and FastSurfer.

Starting with an accurate three-dimensional whole-brain segmentation provided by a deep-learning framework, we perform cortical and sub-cortical reconstruction and fast mapping with an approach that quickly maps the cortex using label fusion functions. It also maps cortical labels and creates a complete FreeSurfer [40]-based brain segmentation alternative. Only 20–30 s of run time contributes to the whole brain segmentation. Hence, SAU-Net has the speed of a supervised deep-learning method with the convenience of a wide range of segmentation-based capabilities offered by conventional neural imaging pipelines.

We extensively validate our proposed segmentation method by assessing segmentation accuracy, qualitative analysis, speed, and test–retest reliability in several publicly available datasets. This is the work of a deep-learning approach that has been thoroughly tested. Even though it is similar to or faster than SOTA deep-learning approaches, SAU-Net increases sensitivity and ability, making it a reliable tool for future large-scale demographic work.

## 2. Materials and Methods

### 2.1. Subjects

We utilized eight brain MRI datasets in our experiments. Six datasets with manual annotations, IXI [41], ADNI [42], HCP [43], PPMI [44], AIBL [45], and the Catholic Aging Brain Imaging (CABI) database [46] of Yeouido St Mary’s Hospital, the Catholic University of Korea, were selected for training and to evaluate segmentation accuracy. The remaining two datasets, CoRR-BNU1 [47] and ChinaSet Multi-Center [48], were used to test the reliability of the segmentation scheme. Table 1 summarizes the number of subjects per dataset, scanner type, matrix size, pixel spacing, the annotated information, and purpose. IXI, ADNI, HCP, PPMI, AIBL, and CABI among our datasets are resampled to a size of 256 × 256 × 256 and pixel spacing of 1.0 × 1.0 × 1.0 (a dataset consisting of IXI, ADNI, HCP, PPMI, AIBL, and CABI is called a “mixed dataset”). Details about the acquisition protocol used for each of the public datasets can be found in their respective references.

Our training data were generated by using FreeSurfer (General Hospital Corporation, Boston, MA, USA, version 6.0) annotations with manual correction in the training datasets for pre-training (250 selected cases from the mixed dataset), and manual annotations (confirmed and corrected by four radiologists) from 48 selected cases of the pre-training dataset for fine-tuning. From the selected 48 cases, 30 cases were used for fine-tuning and 18 cases were only used for testing. For the robust training, the total mixed data are composed of a total of 300 scans which are collected from ten different MRI scanners with 30 scans each, also, the selected 18 scans are collected from nine MRI scanners with two scans each. Table 2 shows the summary of various MRI scanners; the SKYRA scanner was excluded due to failure of the inference process on QuickNAT and FastSurfer (refer to Section 2.5).

### 2.2. Pre-Processing and Augmentation

#### 2.2.1. Pre-Processing

The pre-processing tasks employed in brain image processing include image registration, zero padding, and histogram matching-based intensity normalization. The first step is a rigid registration from the target image to the MNI305 template (256 × 256 × 256) using 3D rigid registration with an Euler transform (SimpleITK) [49,50,51,52]. Next, zero padding is performed: 16 × 16 × 16 padding is used for training and 24 × 24 × 24 is used for testing. The intensities of the obtained scans vary across different devices and can even vary across different scans from the same device. Therefore, to further normalize the intensities across different scans, a histogram matching-based intensity normalization method is employed [53]. In this process, an MRI brain image with a protocol corresponding to a patients’ population are given as input. In the pre-training step, histogram transformation landmark parameters are first learned from MNI-registered image data (250 cases). The images are then transformed using the parameters learned during pre-training. This transformation is image dependent and needs to be done for each given image. All the pre-processing steps were implemented using TorchIO [54].

#### 2.2.2. Data Augmentation for Training

To increase the robustness of the segmentation model, input images for pre-training and fine-tuning were randomly processed using the 3D data augmentation module (“Transforms”) of TorchIO [54] with the following augmentation types and settings:Gaussian noise: Zero mean noise for standard deviation randomly chosen from the range 0 to 0.2Bias field: Uniformly distributed in the range −0.5 to 0.5Affine transform: A scale factor range of 0.9 to 1.1, rotation degree range of −10 to 10, trilinear interpolation (shearing and translation were not applied)Elastic deformation: A probabilistic spin was given to the basic affine transform. Since the affine transform is linear, the image is uniformly deformed as linearity. On the other hand, elastic deformation deforms the image in different directions for each pixel. It is also common in medical imaging data since these are data observed in living things. [55] The parameter is pre-defined as seven control points for each axis X, Y, and Z.

### 2.3. SAU-Net Architecture

The architecture of SAU-Net is largely based on an encoder/decoder which has a skip pathway with a pyramid level similar to that of the U-Net++ architecture with an encoder sub-layer or backbone followed by a decoder sub-layer, as shown in Figure 1a [24,37]. The last layer is a classifier block using softmax. The proposed SAU-Net architecture includes re-designed skip pathways between all encoder and decoder blocks with the same spatial resolution [37]. Each convolution layer is constructed using Evolving Normalization–Activation Layers (EvoNorm) [38] and 3D convolution (Figure 1b). The number of convolution layers depends on the pyramid level, and a 3D ResNeSt block [36] is used at the output of the skip pathways. Essentially, the EvoNorm-based convolution and 3D ResNeSt blocks bring the semantic level of the encoder feature maps closer to that of the corresponding feature maps in the decoder. Based on this architecture, EvoNorm with group normalization is used in the convolution layer of the skip pathways instead of Batch Normalization (BN) and Rectified Linear Unit (ReLU) activation fucntion, and a 3D ResNeSt block is used in the skip connections instead of the existing 3D convolution layer [56]. This process is convenient for improving training efficiency with small batch size and segmentation accuracy for small subcortical structures.

#### 2.3.1. ResNeSt Block with Split-Attention

In the proposed architecture, the 3D ResNeSt block is used in the skip connections to efficiently overcome the disadvantages of small batch sizes and enhance the response of the feature map. Its architecture is shown in Figure 2a. The split-attention module inside the ResNeSt block is a computational unit consisting of a group of feature maps and split-attention operations [36]. Figure 2b presents an overview of a split-attention module.

Groups of feature maps: The feature can be divided into four groups, and the number of groups of feature maps is given by a cardinality K = 2 and a radix R = 2 [36,57]. The total number of feature groups G is four (G = KR).ResNeSt block (Figure 2a): The representations of a cardinal group are concatenated along the channel dimension. As in existing residual blocks, the split-attention module’s final output is produced by a shortcut connection. For blocks with a stride, an appropriate transformation is applied to the shortcut connection to align the output shapes.Split-attention (Figure 2b): An integrated representation for each cardinal group can be obtained by combining it via an element-wise summation across multiple split layers [58,59]. Global semantic information with channel-wise features can be gathered by global average pooling across spatial dimensions [58,59]. A weighted combination of the cardinal group representation is aggregated using channel-wise softmax, where each feature map channel is produced using a weighted combination over split features.

#### 2.3.2. Skip Pathways with Pyramid Levels

Each skip pathway among the nodes consists of a convolution block concatenated with three convolution and ResNeSt block layers. A convolution layer is preceded by a concatenation layer that combines the output from the previous convolution layer of the same concatenation block with the corresponding up-sampled output layer of the lower dense block [37]. Formally, we formulate the split-attention module-based skip pathway as follows:(1)$$x^{p,\;q}=\{\begin{array}{c} C\left(x^{p-i,\;q}\right),\;\;\;\;\;\;\;\;\;\;\;\;\;\;\;\;\;\;\;\;\;\;\;\;\;\;\;\;\;\;\;\;\;\;\;\;\;\;q=0\\ R\left([{\left[x^{p,k}\right]}_{k=0}^{q=1},\;U\left(x^{p+i,q-1}\right)\right]),\;\;\;\;\;\;\;\;\;q>0\; \end{array}\;\;\;\;$$
where function C(·) is a convolution operation followed by an EvoNorm activation function, U(·) denotes an up-sampling layer, R(·) denotes a ResNeSt block, and [∙] denotes the layer with the concatenation function. All prior feature responses accumulate and reach the current node because we make use of a concatenated convolution block along each skip pathway. Finally, the ResNeSt block is applied to the concatenated convolution block as an input. Figure 1a illustrates this structure, showing how the feature maps travel through the top skip pathway of U-Net++.

#### 2.3.3. Classifier Block

The feature map output by the final ResNeSt block is passed to the classifier block, which is a convolutional layer with a 1 × 1 × 1 kernel size that maps the input to an N channel feature map, where N is the number of classes (34 structure labels with non-labels (zero value) in our experiments).

### 2.4. Training with Limited Annotated Data

Our model produces a segmentation result by taking 96 × 96 × 96 patches randomly from the image. Because the 3D MRI volume would not fit into GPU memory using standard deep-learning networks, we randomly employed 128 independent instances of our model as 3D patches to cover the entire MNI space. Our model was a sub-network, whose resolution is a compromise between memory limitations and spatial resolution. For each instance of our model, we modified its decoder part to be compatible with 33-label output. As shown in Figure 1, 33 3D output channels have been employed in each classifier layer.

Although the amount of unlabeled data is rapidly growing, access to labeled data is still limited because of the intense effort needed for manual annotation. Simultaneously, deep-learning success is mainly led by supervised learning, whereas unsupervised approaches still need much research. In our proposed approach, we use the two-step training procedure shown in Figure 3. The first step is pre-training on large unlabeled datasets with auxiliary labels. In this step, we use a large neuroimaging dataset (public dataset) and process it with an existing tool to create auxiliary labels. We employ the widely used FreeSurfer to obtain auxiliary segmentations, but other tools could be used, depending on the application. We pre-train on this large dataset with auxiliary labels, which results in a network that imitates the FreeSurfer segmentations. Pre-training enforces a strong prior on the network, in which robustness to data heterogeneity is encouraged by the diversity of the selected cases. The next step is fine-tuning with a limited amount of manually labeled data: in this step, we take the pre-trained model and fine-tune it with small amount of data that have manual annotations. Instead of learning all filters from scratch, fine-tuning only focuses on rectifying the discrepancies between the auxiliary and manual labels.

### 2.5. Aggregation

When separating the entire MNI space into randomly selected 128 subspaces, if the subspaces overlap (12 × 12 × 12), this provides more than one inference result for a single voxel. A step beyond concatenation is required to obtain the final segmentation label for that voxel from multiple candidates. In this work, the “OR logical label fusion method” was employed to obtain the final segmentation results. Briefly, this method was used to fuse all the segmentations from cropped sub-images according to the size of patches and subspaces overlapping into a single final segmentation in MNI space. The space outside each patch was excluded from the label fusion. The final segmentation in the original target image space was then achieved by registering the segmentation to the original space using a rigid registration. Note that if the segments do not overlap, a naive concatenation is employed to obtain a final single segmentation directly without using label fusion.

### 2.6. Experimental Settings

We evaluated SAU-Net in a series of six experiments to evaluate its accuracy, quality, speed, reproducibility, and sensitivity on a variety of neuroimaging datasets, as summarized in Table 3. In all experiments, SAU-Net was trained as described in Section 2. In the experiment of segmentation performance, we evaluate our segmentation robustness by comparison with other methods. Next, we then performed another three experiments to validate the reliability and consistency of SAU-Net segmentations. To test reliability, we constructed the test dataset as follows:

For a fair comparison, the 30 scans for fine-tuning and the 18 scans for testing in the mixed dataset (Table 1) were consistent with the label definitions of QuickNAT (22 structure labels). Additionally, the existing U-Net [24] and U-Net++ [37], which are well-known architectures, were implemented in the NiftyNet library (based on TensorFlow), which is well optimized for 3D patch-based data loaders and training architectures [60]. The source code and inference model of QuickNAT were downloaded from the author’s GitHub page. [61] Also, the following link is the GitHub page of FastSurfer. [62]

### 2.7. Metrics for Evaluation

We utilize the Dice overlap [63] and average symmetric surface distance (ASSD) score [64] as an accuracy metric of segmentation, $CV_{s\_avg}$ [65] and $CV_{t}$ [65] are used as reliability metrics of segmentation.

Dice overlap score: A statistical index which measures the similarity between the segmented region ($R_{o}$) and ground truth ($R_{G}$). The higher the estimate, the better the estimator. This score has become arguably the most commonly used index in the evaluation of image segmentation, as follows:
(2)$$Dice\;overlap=\frac{2\left|R_{o}\cap^{\;}R_{G}\right|}{\left(\left|R_{o}\right|+\left|R_{G}\right|\right)}$$

Average symmetric surface distance (ASSD): The average of the distances from every point on the boundary ($B_{o}$) of the segmented region to the boundary ($B_{G}$) of the ground truth. The lower the estimate, the better the estimator.
(3)$$ASSD=\frac{1}{\left|B_{o}\right|+\left|B_{G}\right|}\left(\underset{x\in B_{o}}{\sum }d(x,\;B_{G})+\underset{y\in B_{G}}{\sum }d(y,B_{o})\right)$$where d is the Euclidean distance [66].

$CV_{s\_avg}$ (Average intra-session coefficient of variation): The measurement of n total average variance for volume estimation was performed between two sessions of one paired scan set. The $CV_{s\_avg}$ provides the extent of variability in the mean value. The average intra-session coefficient of variation is computed as:
(4)$$\sigma_{s}=\sqrt{\frac{\sum_{i=2}{\left(x_{i}-\mu_{s}\right)}^{2}}{2}}$$
(5)$${\mathrm{CV}}_{s\_\mathrm{avg}}=\frac{(\sum_{i=0}^{n}\sigma_{s}/\mu_{s})\;}{n}\times 100$$where $x_{i}$ and $\mu_{s}$ are the ith session measurements and mean of each pair of scan sets.

$CV_{t}$ (total coefficient of variation): The coefficient of variation in volume estimates within the total variance over n scans per subject. The $CV_{t}$ is computed as:
(6)$$\sigma_{t}=\sqrt{\frac{\sum_{i=n}{\left(x_{i}-\mu_{t}\right)}^{2}}{n}}$$
(7)$${\mathrm{CV}}_{t}=\frac{\sigma_{t}}{\mu_{t}}\times 100$$where $x_{i}$ and $\mu_{t}$ are the ith measurements of scans and mean of total scans per subject.

### 2.8. Statistical Analysis

Accuracy: We measured using Dice overlap and ASSD for the selected 18 mixed datasets and used the mean and standard deviation for evaluation.Speed: The atlas-based methods brought the results of the QuickNAT, and the rest of the methods measured the runtime of the final segmentation result after the pre-processing, inference, and post-processing were completed.Device inter-variability: In Table 2, we show the acquired MRI scans according to ten MRI scanners, and conducted training and evaluation. These are necessary for robust training, and it is essential to show consistent performance for different devices. We evaluated the device inter-variability by comparing the overall mean Dice overlap and standard deviation.Intra-session variability: This dataset was released to test the reliability of automated whole brain segmentation algorithms in estimating volumes of brain regions (excluding the corpus callosum) with 32 structure labels using a test–retest protocol. In this dataset, 47 MRI T1 scan pairs were taken on the same device with an interval of time between each scan (that is, two scans per session). All the scans were acquired over a period of six weeks (40.94 ± 4.51 days). We analyzed the $CV_{s\_avg}$ of the overall 47-pair scan set. Ideally, as volume atrophy is almost insignificant within a period of 30 days, the coefficient of variation in the estimates should be zero.Multi-center variability: In this experiment, we validated the reliability and robustness of estimating volumes across scans acquired from multiple centers. To do this, we used images from the ChinaSet Multi-Center dataset. This dataset includes scans from three patients who traveled to 12 different medical centers in China. Each of the 12 imaging centers used MRI scanners manufactured by the same vendors (different scanning parameters). We analyzed the $CV_{t}$ of the total variance over 12 scans per subject.

### 2.9. Training Setup

The SAU-Net was implemented using TorchIO library on the PyTorch 1.4 version and trained using Cross Entropy loss function [67] and AdamW optimizer with AMSGrad [68,69]. The learning rate of pre-training was 0.001; that of fine-tuning was 0.0001. The training was conducted for 300,000 iterations. The batch size was set to four, which was the limit that could be handled by the 48 GB RAM of the two QUADRO RTX6000 GPUs.

## 3. Results

### 3.1. Results of Segmentation Performance

#### 3.1.1. Accuracy

In this experiment, we compared the performance of SAU-Net with SOTA methods and evaluated the impact of pre-training and fine-tuning. Table 4 shows the results measured using Dice overlap scores and the average symmetric surface distance (ASSD) with 22 structure labels.

Along each row, we observed that for our models, the fine-tuned model yields almost significantly (*p* < 0.05) better performance than the model pre-trained using the FreeSurfer labels. Table 4 and Figure 4 and Figure 5 show the Dice overlap score and ASSD results for comparison with SOTA brain segmentation methods. Along each row, we observe that for all models, the fine-tuned SAU-Net model yields better performance than U-Net (a difference of 0.5% in Dice score and 0.124 mm in ASSD), FastSurfer (1.3%, 0.131 mm), U-Net++ (1.3%, 0.151 mm), and QuickNAT (5.2%, 0.23 mm).

Additionally, we performed the two-sided t-test between SAU-Net fine-tuned model and the other each model. However, except for SAU-Net pre-trained model since it was trained using the same dataset. The U-Net, U-Net++, and QuickNAT show statistical significance in most of the whole brain ROI regions, but FastSurfer is challenging to confirm statistical significance in the Dice overlap results. (p>0.05 in more than half of the whole brain ROI regions) All the models show statistical significance in most of the whole brain ROI regions in the ASSD score results.

#### 3.1.2. Qualitative Analysis

Sample segmentation results are shown in Figure 6 and Figure 7 for U-Net, U-Net++, FastSurfer, QuickNAT, pre-trained SAU-Net, and fine-tuned SAU-Net along with the radiologist-confirmed FreeSurfer labels. We indicate the important subcortical structures, the left (blue) lateral ventricles, with yellow boxes. We can observe the under-inclusion of the left lateral ventricle in the results obtained by U-Net, U-Net++, and FastSurfer. We also often observe specific classified regions where aliasing appears in the results of U-Net, QuickNAT, and FastSurfer but not in the results of U-Net++ and SAU-Net (indicated by red dashed circles).

#### 3.1.3. Speed

Conventional template-based whole-brain segmentation approaches implemented for 3D deformable volume registration are slow [70]. The results take a long time because a single pair-wise registration takes about 2 h. In the Multi-Atlas Labeling Challenge (MALC) [41] dataset, the approximate segmentation runtime for Penn Image Computing & Science Lab (PICSL) [15] software for multi-atlas segmentation is 30 h per volume. FreeSurfer has its own template and takes about 4–6 h per volume. FastSurfer and QuickNAT are implemented using a 2D multi-view-based segmentation approach, and they take about 50 and 25 s, respectively, to run. Additionally, the 3D patch-based U-Net and U-Net++ take around 18–27 s to run [60]. These methods are substantially faster than the deformable registration-based segmentation approach. In contrast, SAU-Net segments a volume in 28 s, which is comparable to the runtimes of the faster methods. We present the segmentation runtimes in Figure 8 on a logarithmic scale.

### 3.2. Results of Segmentation Reliability

#### 3.2.1. Device Inter-Variability

In the Dice overlap results, the fine-tuned SAU-Net model outperforms other models in about half of the eight MRI devices included in the dataset in this study (Table 2 and Table 5). In addition, SAU-Net outperforms other models because it has the highest average Dice score (fine-tuned SAU-Net) and lowest standard deviation (pre-trained SAU-Net) on all devices.

#### 3.2.2. Intra-Session Reliability: CoRR-BNU1

We analyzed the $CV_{s\_avg}$ of the overall 47-pair scan set of CoRR-BNU1. The lower the estimate, the better the estimator. We compare SAU-Net with FastSurfer in this regard. The results of the experiment are reported in Table 6. In the result, fine-tuned SAU-Net performs better than the FastSurfer model (except for accumbens right).

#### 3.2.3. Multi-Center Reliability: ChinaSet Multi-Center

We analyzed the $CV_{t}$ of the total variance over 12 scans per subject (Table 7). We compared SAU-Net with FastSurfer and report the results in Table 7. SAU-Net is more robust for the cerebral white matter (WM) (left/right), lateral ventricle (left/right), caudate (left), accumbens (left), putamen (left/right), amygdala (left), and thalamus (left) structures, whereas FastSurfer performs better on the caudate (right), accumbens (right), amygdala (right), hippocampus (left/right), pallidum (left/right), and thalamus (right) structures. Overall, this challenging experiment demonstrates that SAU-Net and FastSurfer are almost equally robust.

## 4. Discussion

### 4.1. Comparison with Deep-Learning Methods

In recent years, CNNs have been proposed for whole-brain segmentation. [26,27,28,29,30,31,32,34,35,71,72] QuickNAT yields lower accuracy than other approaches while requiring only 20–30 s to run. QuickNAT has low accuracy because this model was pre-trained with FreeSurfer labels, but fine-tuned with Neuromorphometrics Inc.’s labels [73], so the overall accuracy is low. FastSurfer is a network proposed for segmenting the structure labels of FreeSurfer based on a 2D multi-view framework, which yields higher accuracy than QuickNAT. Although SLANT could not be directly compared with the SAU-Net model because it has more segmentation labels than our model, the performance seems to have similar or lower accuracy to SAU-Net. However, SLANT is slower in terms of speed (10–15 min). We compared U-Net, U-Net++, QuickNAT, and FastSurfer using an identical experimental setup with the Dice overlap and ASSD metrics. By evaluating SAU-Net on six different datasets and performing a reliability study on two datasets, we have presented a comprehensive evaluation of current CNN-based methods for brain segmentation.

### 4.2. Pre-Training with Auxiliary Labels

Even if the deep learning-based methods have been highly effective, their cost is high and they require an extensive amount of annotated data for robust training [39]. Access to plentiful annotated data for training is challenging for medical-based applications because of the high cost of creating expert manual annotations. This issue is more critical for deep learning-based methods, where each image slice corresponds to a one-shot data representation, in contrast to patch-based approaches, where many patches can be extracted from a volume [28]. We evaluated a training strategy that leverages extensive unlabeled annotation data and a small amount of manual annotation data to train our SAU-Net model to overcome this issue effectively [29]. We utilized FreeSurfer to automatically segment brain structure labels from unlabeled data, and these labels were used as auxiliary labels to pre-train our model. This pre-trained model, which imitates FreeSurfer’s segmentations, is then fine-tuned with a small amount of manually labeled data to obtain the final model. Our results have demonstrated that a model trained with this training strategy outperforms the same model pre-trained on auxiliary label data only.

### 4.3. Architectural Design for Deep Learning

The architecture of SAU-Net has been modified to address the challenges associated with multi-label brain segmentation. Our architecture offers faster processing and a broader context than patch-based U-Net and U-Net++. This is because, by adding the split-attention module to the U-Net++ structure, the separation between labels seems to be improved thanks to the split grouping of the feature channels [36]. In the skip pathways with the pyramid levels, concatenating all the layers, which enforces spatial consistency, is an essential aspect for segmenting small subcortical structures. In addition, using EvoNorm [38] instead of a combination of ReLU and batch normalization overcomes the disadvantage that large batch sizes cannot be used because of the property of 3D patches, and shows that sufficient performance is achieved even when small batch sizes are used in training. Our results show that the SAU-Net architecture is a significant improvement on the architectures of the SOTA deep-learning models.

### 4.4. Segmentation Performance

We demonstrated the high accuracy of SAU-Net in a comprehensive experiment that covers variations in acquisition parameters and neuroanatomy. In an experiment on the mixed dataset, we demonstrated that SAU-Net provides segmentation accuracy and ASSD scores that are slightly higher than those of the SOTA deep-learning-based methods (Table 3, Figure 4 and Figure 5). Our results are similar or slightly better than the performance (for 22 structure labels) of all other methods except for QuickNAT. However, our models show better results in the subcortical areas such as the lateral ventricle. Notably, SAU-Net has not failed on any scan in our datasets.

### 4.5. Segmentation Reliability

In two additional experiments, we evaluated the reliability of SAU-Net. We observed high consistency for the segmentation of nine representative structures (left and right) on test–retest data with less than 1–4% variation in most brain structures (Table 5). The test–retest data were acquired in the same scanner. We extended the evaluation to a more challenging dataset, where the same subject was scanned in the same machines with various scanning parameters at different sites. As expected, the variation increased in this setup, but the reliability was comparable to that of FastSurfer (Table 6). Since FastSurfer presented the best performance amongst SOTA deep-learning methods (except for NiftyNet-based U-Net style architecture [60]), we have conducted further analysis to compare the reliability of SAU-Net.

However, our results showed that the more unconstrained, deep-learning-based segmentation could achieve higher reliability.

### 4.6. Limitations

As with any brain segmentation method, SAU-Net has limitations. Our mixed dataset includes various public and patient data from domestic hospitals. However, our labeled data are an improved manual annotation based on FreeSurfer’s segmentation, so it cannot be directly compared with FreeSurfer. In the future, we need to utilize better-defined manual annotations. Additionally, the SAU-Net model, like other SOTA methods, has lower accuracy in smaller subcortical regions such as the accumbens and amygdala than in other brain regions. The accuracy was improved after the model was fine-tuned, however, this issue remains a challenging task.

## 5. Conclusions

We introduced SAU-Net, a deep-learning-based method for brain segmentation that achieved a great performance with respect to existing methods and is as fast as or orders of magnitude faster than the SOTA deep-learning and template-based approaches. We have demonstrated that SAU-Net generalizes well to other unseen datasets when the training data differ from the testing data and yields high segmentation accuracy. This high segmentation accuracy enhances group analyses by obtaining effect sizes and significance values that better match those of manual segmentations. Furthermore, because of its high test–retest accuracy, it can be effectively used for longitudinal studies. We believe SAU-Net can be a highly impactful segmentation network due to its robustness to neuroanatomical variability, allowing for almost instantaneous access to accurate imaging biomarkers along with its fast processing time.

## Figures and Tables

**Figure 1 brainsci-10-00974-f001:**
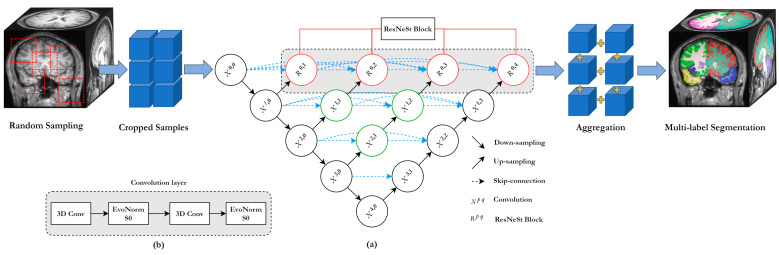
Split-attention U-Net (SAU-Net): (**a**) overall architecture and (**b**) modified convolution layer. (EvoNorm-S0: Evolving normalization layer with the Group Normalization and Swish activation function).

**Figure 2 brainsci-10-00974-f002:**
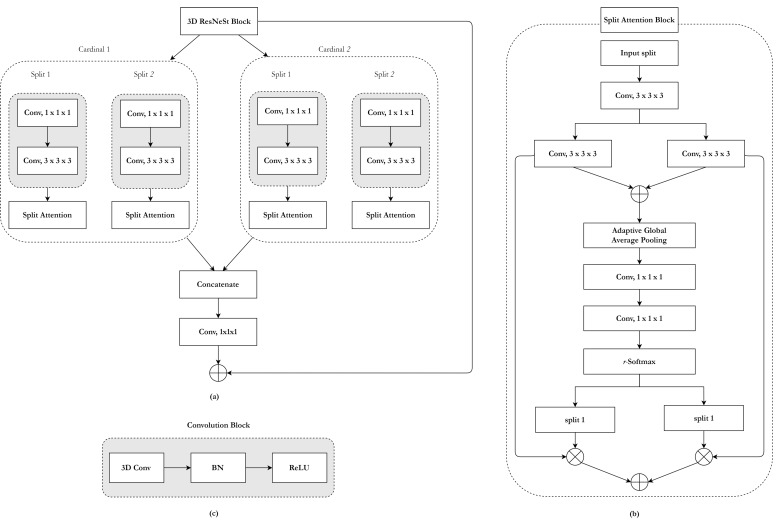
ResNeSt block: (**a**) overall architecture, (**b**) split-attention module, and (**c**) convolution layer. (BN: Batch Normalization, ReLU: Rectified Linear Unit activation function).

**Figure 3 brainsci-10-00974-f003:**
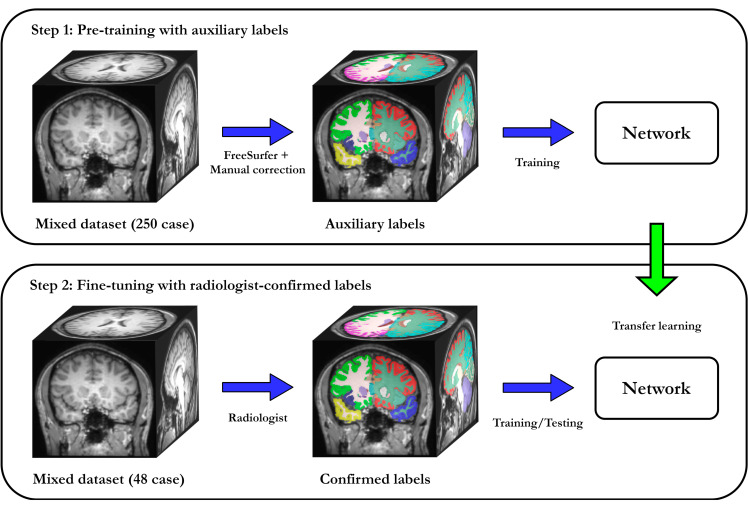
Two-step training strategy for SAU-Net. First, we used FreeSurfer to automatically segment a large unlabeled dataset (a mixed dataset consisting of 250 scans listed in the Table 1). These labels are referred to as auxiliary labels and used for pre-training. Second, we fine-tuned the network on 48 scans selected from the same public dataset that were manually annotated by four radiologists. Fine-tuning does not start from scratch but continues to optimize the pre-trained model to leverage the scarce amount of data with manual annotations.

**Figure 4 brainsci-10-00974-f004:**
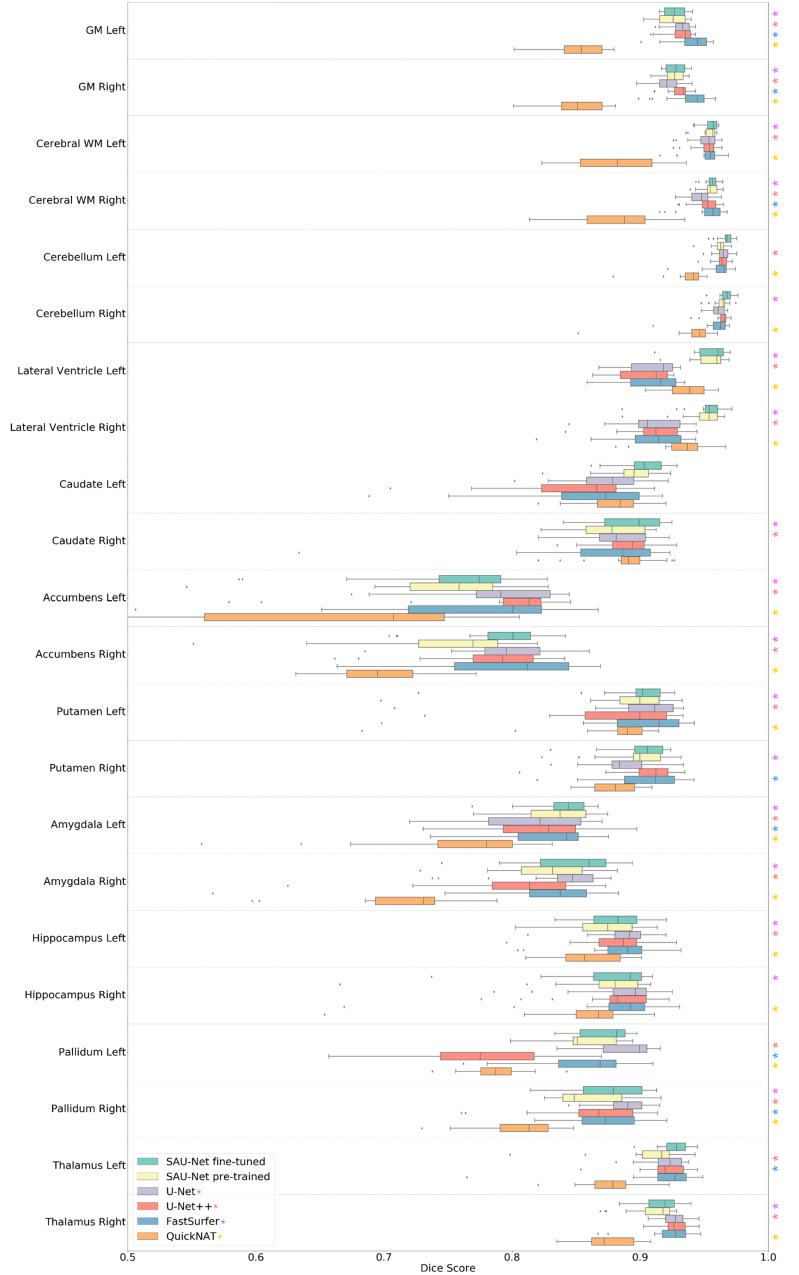
Comparison of the Dice overlap score results of SAU-Net, U-Net, U-Net++, FastSurfer, and QuickNAT on the 18 test cases from the mixed dataset for all 22 structure labels. Statistical significance (*p* < 0.05) with respect to SAU-Net fine-tuned model is presented by each colored asterisk. The *p*-values were estimated using a two-sided t-test. The box represents the confidence interval with 50th percentile, and the line outside the box is 95th percentile, while the whiskers represent the rest of the data except for data it determines to be outliers. (GM: Gray Matter, WM: White Matter).

**Figure 5 brainsci-10-00974-f005:**
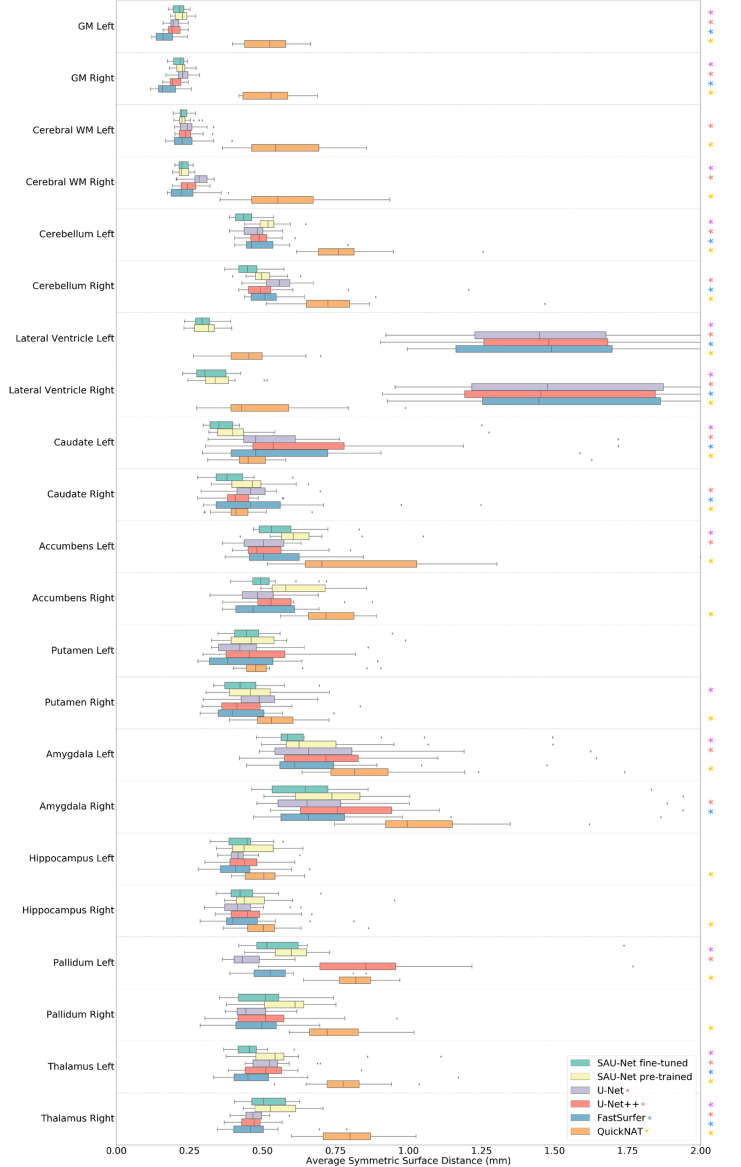
Comparison of ASSD results of SAU-Net, U-Net, U-Net++, FastSurfer, and QuickNAT on the 18 test cases from the mixed dataset for all 22 structure labels. Statistical significance (*p* < 0.05) with respect to SAU-Net fine-tuned model is presented by each colored asterisk. The *p*-values were estimated using a two-sided t-test. The box represents the confidence interval with 50th percentile, and the line outside the box is 95th percentile, while the whiskers represent the rest of the data except for data it determines to be outliers. (GM: Gray Matter, WM: White Matter).

**Figure 6 brainsci-10-00974-f006:**
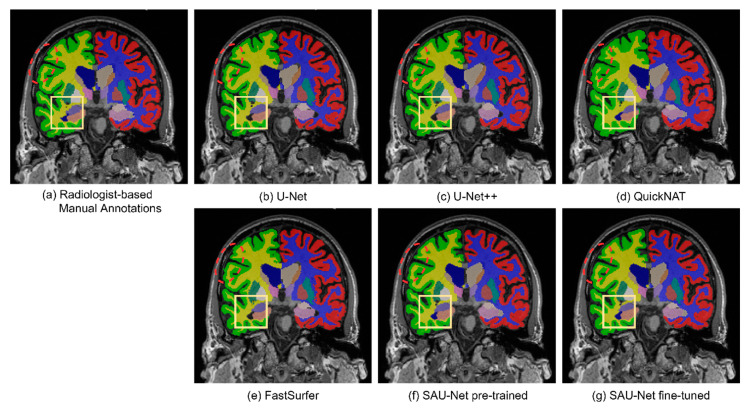
Example results for qualitative evaluation: (**a**) manually annotated image, (**b**) U-Net, (**c**) U-Net++, (**d**) QuickNAT, (**e**) FastSurfer, (**f**) pre-trained SAU-Net, and (**g**) fine-tuned SAU-Net.

**Figure 7 brainsci-10-00974-f007:**
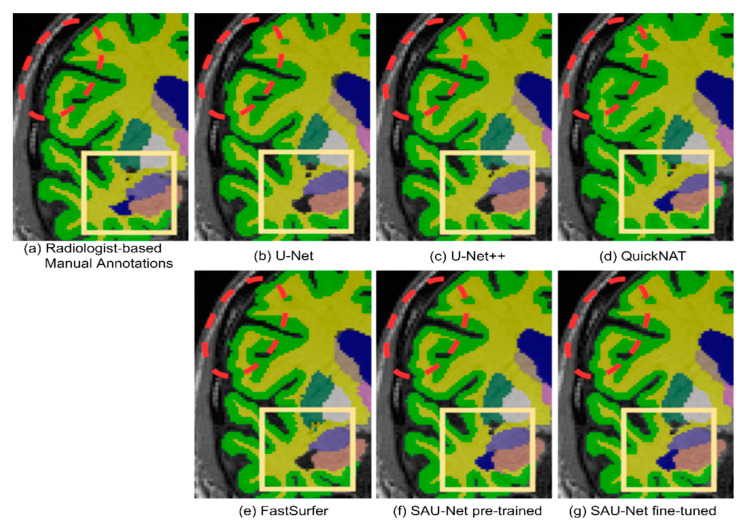
Example results with zoom-in for qualitative evaluation: (**a**) manually annotated image, (**b**) U-Net, (**c**) U-Net++, (**d**) QuickNAT, (**e**) FastSurfer, (**f**) pre-trained SAU-Net, and (**g**) fine-tuned SAU-Net. the red dashed circles indicate low-quality results, such as aliasing, and the yellow boxes show under-inclusion of the left lateral ventricles (blue color).

**Figure 8 brainsci-10-00974-f008:**
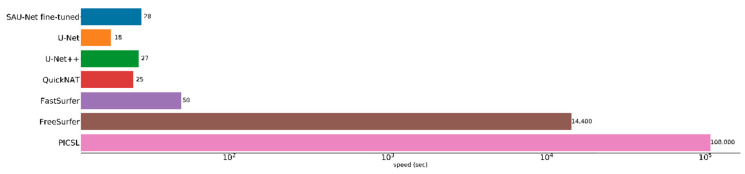
Runtime performance of the comparison methods.

**Table 1 brainsci-10-00974-t001:** Summary of the public datasets used for training and testing.

Dataset	No. of Subjects	Scanner Type	Matrix Size	Pixel Spacing (mm)	Annotations	Purpose
IXI	17	T1	256 × 256 × 256	1.0 × 1.0 × 1.0	33 labels	All
ADNI	207	“	“	“	“	“
HCP	30	“	“	“	“	“
PPMI	8	“	“	“	“	“
AIBL	14	“	“	“	“	“
CABI	24	“	“	“	“	“
Total mixed dataset	300	“	“	“	“	“
CoRR-BNU1	47	“	144 × 256 × 256	1.3 × 1.0 × 1.0	None	Only testing
ChinaSet Multi-Center	3	“	240 × 256 × 256	1.0 × 1.0 × 1.2	None	Only testing
Total testing only dataset	50					

33 labels: 33 structure labels of the whole brain. All: training/testing. “: indication that it is the same as the row line above.

**Table 2 brainsci-10-00974-t002:** Summary of the numbers of subjects according to 10 magnetic resonance imaging (MRI) scanners and description.

MRI Scanner	No. of Subjects	Pre-Training	Fine-Tuning	Testing	Description
Intera	30	25	3	2	Philips Intera 3.0T
Prisma fit	“	“	“	“	Siemens 3.0T Magnetom Prisma Fit
Ingenia	“	“	“	“	Philips MRI Ingenia 3.0T
Signa HDxt	“	“	“	“	GE Signa HDxt 3.0T MR system
Signa HDx	“	“	“	“	GE Signa HDx 3.0T MRI system
Achieva	“	“	“	“	Philips Achieva 3.0T MRI system
DISCOVERY MR750	“	“	“	“	GE Discovery™ MR750 3.0T system
TrioTim	“	“	“	“	Siemens 3.0T Trio TIM MRI scanner
Verio	“	“	“	“	Siemens 3.0T MAGNETOM Verio
SKYRA	“	“	“		Siemens 3.0T MAGNETOM Skyra

“: indication that it is the same as the row line above.

**Table 3 brainsci-10-00974-t003:** Experimental settings for the evaluation of SAU-Net.

Evaluation	Experiment	Testing	Purpose of Experiment
Segmentation Performance	Accuracy	Selected mixed dataset for testing only (18) *	Comparison of Dice overlap and ASSD scores with other methods
	Qualitative Analysis	“	Comparison of segmentation quality with other methods
	Speed	“	Comparison of runtime speed with other methods
Segmentation Reliability	Device Inter-Variability	“	Inter-scan reliability with nine MRI scanners
	Intra-Session Reliability	CoRR-BNU1 (47)	Intra-session reliability with two scans per session (same scanner)
	Multi-Center Reliability	ChinaSet Multi-Center (3)	Reliability across 12 locations with same scanners

ASSD: average symmetric surface distance. * Numbers in parentheses: numbers of the corresponding dataset. “: indication that it is the same as the row line above.

**Table 4 brainsci-10-00974-t004:** Comparison of the Dice overlap score and ASSD of SAU-Net with 22 structure labels (mean ± standard deviation).

Method	Dice overlap	ASSD (mm)
U-Net	0.892 ± 0.049 *****	0.557 ± 0.326 *****
U-Net++	0.884 ± 0.061 *****	0.584 ± 0.334 *****
QuickNAT	0.845 ± 0.080 *****	0.663 ± 0.167 *****
FastSurfer	0.884 ± 0.062	0.564 ± 0.332 *****
SAU-Net pre-trained	0.887 ± 0.064	0.479 ± 0.159
SAU-Net fine-tuned	0.897 ± 0.056	0.433 ± 0.136

Star symbols in bold (*****): Statistical significance (*p* < 0.05) between the SAU-Net fine-tuned model (reference) and the other models. (Except for SAU-Net pre-trained model since the training data is same with the SAU-Net fine-tuned model).

**Table 5 brainsci-10-00974-t005:** Device inter-variability with respect to the Dice overlap of SAU-Net and other State-Of-The-Art (SOTA) methods for 22 structure labels (mean ± standard deviation).

MRI Scanner	SAU-Net Fine-Tuned	SAU-Net Pre-Trained	U-Net	U-Net++	FastSurfer	QuickNAT
Intera	0.907 ± 0.055	0.892 ± 0.077	0.901 ± 0.055	0.894 ± 0.068	0.904 ± 0.073	0.851 ± 0.103
Prisma fit	0.891 ± 0.073	0.898 ± 0.051	0.900 ± 0.039	0.898 ± 0.048	0.902 ± 0.049	0.846 ± 0.076
Ingenia	0.903 ± 0.054	0.877 ± 0.129	0.878 ± 0.121	0.877 ± 0.111	0.876 ± 0.129	0.837 ± 0.134
Signa HDxt	0.884 ± 0.110	0.883 ± 0.064	0.890 ± 0.055	0.882 ± 0.062	0.875 ± 0.073	0.842 ± 0.085
Signa HDx	0.894 ± 0.053	0.875 ± 0.100	0.872 ± 0.100	0.897 ± 0.055	0.910 ± 0.044	0.857 ± 0.069
Achieva	0.917 ± 0.044	0.901 ± 0.058	0.900 ± 0.055	0.895 ± 0.057	0.901 ± 0.058	0.852 ± 0.079
DISCOVERY MR750	0.899 ± 0.056	0.882 ± 0.072	0.883 ± 0.064	0.873 ± 0.081	0.885 ± 0.076	0.837 ± 0.102
TrioTim	0.899 ± 0.061	0.891 ± 0.061	0.900 ± 0.047	0.885 ± 0.07	0.896 ± 0.054	0.860 ± 0.069
Verio	0.898 ± 0.068	0.885 ± 0.076	0.888 ± 0.064	0.879 ± 0.073	0.880 ± 0.074	0.877 ± 0.074
Total	0.897 ± 0.056	0.887 ± 0.064	0.892 ± 0.049	0.884 ± 0.049	0.884 ± 0.062	0.845 ± 0.080

**Table 6 brainsci-10-00974-t006:** Variation in volume measurement per structure. Fine-tuned SAU-Net is compared with FastSurfer in terms of the $CV_{s\_avg}$ between them. The mean volume estimates per structure are also reported.

Structure	Volume (cc)		$CV_{s\_avg}\;(\%)$	
	SAU-Net	FastSurfer	SAU-Net	FastSurfer
Cerebral WM Left	224.703	238.169	0.294	2.553
Cerebral WM Right	231.598	238.053	0.357	2.496
Lateral Ventricle Left	7.158	6.632	1.737	6.509
Lateral Ventricle Right	6.716	5.534	1.688	6.622
Caudate Left	3.969	3.839	1.026	3.291
Caudate Right	3.457	3.996	1.196	3.551
Accumbens Left	0.559	0.66	3.905	3.599
Accumbens Right	0.668	0.654	2.793	4.37
Putamen Left	5.697	5.514	1.104	2.537
Putamen Right	5.726	5.579	0.832	2.479
Amygdala Left	2.049	1.75	1.967	2.827
Amygdala Right	2.02	1.865	1.502	2.872
Hippocampus Left	4.123	4.243	1.132	2.524
Hippocampus Right	4.544	4.421	1.082	2.355
Pallidum Left	2.063	2.174	3.648	2.872
Pallidum Right	1.885	2.124	2.29	2.766
Thalamus Left	8.255	7.972	1.057	1.997
Thalamus Right	7.275	7.855	0.781	1.95

SAU-Net: fine-tuned SAU-Net model.

**Table 7 brainsci-10-00974-t007:** $CV_{t}$ in volume estimation for the nine representative structures (left/right) for each subject, using fine-tuned SAU-Net (SAU-Net) and FastSurfer.

Structure	$\mathrm{Subject}\;\mathrm{ID}\;\mathrm{with}\;CV_{t}\;(\%)$					
	Subject 1		Subject 2		Subject 3	
	SAU-Net	FastSurfer	SAU-Net	FastSurfer	SAU-Net	FastSurfer
Cerebral WM Left	1.126	1.32	1.979	2.354	1.887	2.105
Cerebral WM Right	0.885	1.235	1.554	2.018	2.004	2.286
Lateral Ventricle Left	4.965	4.776	2.812	3.088	4.151	5.213
Lateral Ventricle Right	5.571	5.978	2.106	3.239	3.639	4.215
Caudate Left	1.601	1.789	1.807	2.674	2.323	1.366
Caudate Right	1.782	1.603	1.39	1.081	1.262	1.742
Accumbens Left	7.341	12.679	4.492	7.089	5.828	4.597
Accumbens Right	8.481	8.024	7.64	6.688	8.72	7.082
Putamen Left	2.636	2.707	1.562	1.401	1.737	2.58
Putamen Right	1.725	2.038	1.082	1.202	1.519	2.042
Amygdala Left	8.943	3.86	2.941	4.843	4.999	5.077
Amygdala Right	11.01	8.474	3.599	3.693	7.719	3.229
Hippocampus Left	3.288	1.669	1.577	1.861	2.273	1.467
Hippocampus Right	3.958	3.64	3.04	2.101	3.553	3.295
Pallidum Left	4.989	4.657	5.515	3.046	7.811	3.57
Pallidum Right	2.86	2.658	3.725	2.961	7.26	5.007
Thalamus Left	0.822	1.085	1.694	1.726	2.217	1.279
Thalamus Right	2.523	2.541	1.785	1.094	3.152	2.438

SAU-Net: fine-tuned SAU-Net model.

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
