# Peer review of "Split-Attention U-Net: A Fully Convolutional Network for Robust Multi-Label Segmentation from Brain MRI"

_brainsci, 2020, doi:10.3390/brainsci10120974_

Round 1

Reviewer 1 Report

In this work, the authors present a convolutional neural network for segmentation of MR images of the brain. The network was trained using six databases of annotated images. The accuracy of the segmentations was evaluated and compared to other, state-of-the-art methods. Also the computational speed was evaluated and compared to the state-of-the-art methods. The authors conclude that the proposed method achieves superior performance compared to existing methods, while being faster than many of state-of-the-art methods.

General comments

While the authors have made a decent job of conducting the experiments, a concern is the novelty of the presented work, where the introduction is more or less identical to the paper by Huo et al. (reference 44). Other parts needing improvements are the methods and results sections, which are difficult to follow and often intermixed. It should further be made more clear what exactly is new, compared to e.g. the U-Net++ method, or the paper by Huo et al. Is the SAU-Net a completely new network, or was it pre-existing but modified? The overall structure of the manuscript is unsatisfying, with presented results that have not been described in the Methods section. The references need to include full author lists and journal names.

Specific points

  1. The introduction, besides being largely taken from the paper by Huo et al., is overly lengthy, and more suited for a review paper. It should preferably be shortened.
  2. Throughout the manuscript: It is not clear whether state-of-the-art (SOTA) refers to existing deep-learning based methods, or only atlas-based, non-deep learning methods.
  3. Page 3, Line 120-121: “Seven datasets…were selected…”. Table 1 lists six annotated datasets. How many subjects were used from each dataset? One or several? It would be nice if Table 1 also included one column with a very brief description of the MRI dataset, e.g. T1, MRI scanner type, and matrix size.
  4. P3, L121: Please include references for the CABI, the CoRR-BNU1, and the ChinaSet-Multi Center datasets.
  5. P4, L147-149: What is meant by “the protocol corresponding to a patients’ population”? Please explain. And which are the “a few additional input parameters”?
  6. Figure 3: Which is the “public dataset consisting of 250 scans”? Is this listed in Table 1? Please add a reference.
  7. P7-9, L268-287 (except Table 3): this text belongs to the Methods section.
  8. Table 2: The readability of this table is poor. Please improve the table design. What are the numbers in parentheses? The number of subjects? Please explain in the legend.
  9. P8, L280: please explain how ASSD is defined and calculated.
  10. P9, L288: “…the fine-tuned model yields almost significantly (p < 0.05) better performance…”. Where are the p-values listed? And “almost significantly” is too vague. Please state the p-value explicitly. P>0.05 is not significant.
  11. Figures 4 and 5 are too small to be readable. Consider splitting them in sub-parts.
  12. P9, L295-6: “…our fine-tuned SAU-Net model outperforms FastSurfer by an average margin (1.5%, 0.131 mm) that is clearly not statistically significant”. I do not understand this sentence. How can SAU-Net be said to outperform FastSurfer, when the margin is “clearly not significant”?
  13. P9, L307: How exactly was “Device Inter-variability” tested? Please describe in the Methods. I also do not understand why a comparison of different scanners is relevant in a manuscript comparing segmentation methods.
  14. Table 4: Use “MRI scanner” instead of “Device”. Why are some numbers bold?
  15. P11, L322: “We can observe the under-inclusion of the left and right lateral ventricle…”. This is, for me, impossible to detect in Figure 6.
  16. P12, Evaluation of Segmentation Reliability: this experiment should be explained in detail in the Methods section, not in the Results. It is also difficult to understand exactly what was calculated.
  17. Tables 5 and 6: why was FastSurfer chosen for the comparison, and not any of the other methods?

Author Response

Response to Reviewer 1 Comments

We are grateful to the reviewer for their time and constructive comments on our manuscript. We have implemented their comments and suggestions and wish to submit a revised version of the manuscript for further consideration in the journal.

Manuscript ID brainsci-992710 entitled "Split-Attention U-Net: A fully convolutional network for robust multi-label segmentation from brain MRI"

Comments and Suggestions for Authors

In this work, the authors present a convolutional neural network for segmentation of MR images of the brain. The network was trained using six databases of annotated images. The accuracy of the segmentations was evaluated and compared to other, state-of-the-art methods. Also, the computational speed was evaluated and compared to the state-of-the-art methods. The authors conclude that the proposed method achieves superior performance compared to existing methods, while being faster than many of state-of-the-art methods.

General comments

While the authors have made a decent job of conducting the experiments, a concern is the novelty of the presented work, where the introduction is more or less identical to the paper by Huo et al. (reference 44). Other parts needing improvements are the methods and results sections, which are difficult to follow and often intermixed. It should further be made more clear what exactly is new, compared to e.g. the U-Net++ method, or the paper by Huo et al. Is the SAU-Net a completely new network, or was it pre-existing but modified? The overall structure of the manuscript is unsatisfying, with presented results that have not been described in the Methods section. The references need to include full author lists and journal names.

Response of General Comments:

We appreciate your careful review and suggestions for the development of our paper. We are grateful for the opportunity to address your concerns in the accompanying amended revision. Details of our response are below each specific comment.

1) We extensively modified the “Introduction” with our method’s rationale

2) We splitted the “Results” to “Metrics for Evaluation”, “Statistical Analysis” and “Experimental Settings” of the “Materials and Methods”

3) The references were modified to full author lists

* of revised manuscript: revised manuscript with all TRACK of revision tab of WORD processor

Specific points

  1. The introduction, besides being largely taken from the paper by Huo et al., is overly lengthy, and more suited for a review paper. It should preferably be shortened.

Response 1: We have largely modified and drastically reduced to revise our introduction as follows.

> Page 3 41 ~ Page 6 190

  1. Throughout the manuscript: It is not clear whether state-of-the-art (SOTA) refers to existing deep-learning based methods, or only atlas-based, non-deep learning methods.

Response 2: We agree with the reviewer’s point of view. We mentioned about state-of-the-art (SOTA) earlier in the Introduction (for atlas-based segmentation), but again mentioned when referring to the related works for deep learning. We modified our manuscript according to reviewer’s comment.

> Page 2 line 59 ~ 60 Remove ‘state-of-the-art (SOTA)’

> Page 11 line 333 Remove ‘SOTA’

  1. Page 3, Line 120-121: “Seven datasets…were selected…”. Table 1 lists six annotated datasets. How many subjects were used from each dataset? One or several? It would be nice if Table 1 also included one column with a very brief description of the MRI dataset, e.g. T1, MRI scanner type, and matrix size.

Response 3: Suggestions are well taken. We modified the paragraph about the Page 3, Line 120-121 and modified detail of Table 1 in our manuscript.

> Page 3 line 120 ~ 121 Seven datasets~ to Six datasets~

> Page 3 line 125 ~ 129 Add more description of dataset (Scanner type, Matrix size, Pixel spacing)

> Page 3 line 130 Modify the Table 1

  1. P3, L121: Please include references for the CABI, the CoRR-BNU1, and the ChinaSet-Multi Center datasets.

Response 4: We modified the reference about the P3, L121.

> Page 3 line 138 ~ 139 Remove the reference 52, 53

> Page 3 line 123 ~ 124 Add the reference 52, 53

  1. P4, L147-149: What is meant by “the protocol corresponding to a patients’ population”? Please explain. And which are the “a few additional input parameters”?

Response 5: As it seemed unclear to convey the meaning, the contents of “the protocol corresponding to a patients’ population” and “a few additional input parameters” were removed, and the paragraph was reorganized.

> Page 4 147 ~ 155 Reorganize the paragraph

  1. Figure 3: Which is the “public dataset consisting of 250 scans”? Is this listed in Table 1? Please add a reference.

Response 6: A public dataset consisting of 250 scans is listed in Table 1. We clearly modified the sentence and added the table reference.

> Page 7 251 Modify the sentence

  1. P7-9, L268-287 (except Table 3): this text belongs to the Methods section.

Response 7: We agree with the reviewer’s point of view. So, we reorganized to the section of ‘Methods’ which is added ‘Experimental Settings’, ‘Metrics for Evaluation’ and ‘Training Setup’.

> Page 7 267 Add ‘Experimental Settings’, ‘Metrics for Evaluation’ and ‘Training Setup’.

> Page 7 222 ~ 226 Move ‘SAU-Net Training’ paragraph to ‘Training Setup’ in the ‘Methods’ section (Page 7 306 ~ 310 of revised manuscript).

> Page 8 281 ~ 288 Move the paragraph to ‘Experimental Settings’ in the ‘Methods’ section (Page 8 276 ~ 280 of revised manuscript).

  1. Table 2: The readability of this table is poor. Please improve the table design. What are the numbers in parentheses? The number of subjects? Please explain in the legend.

Response 8: The numbers in parentheses are the numbers of corresponding dataset (Section 2.1). We redesigned the Table 2 and added the legend of the numbers in parentheses according to the Reviewer’s comments.

> Page 7 275 Redesign the Table 2

> Page 7 276 Modify the legend of Table2

  1. P8, L280: please explain how ASSD is defined and calculated.

Response 9: ASSD (Average Symmetric Surface Distance) is the average of the distances from every points on the boundary ( ) of the segmented region to the boundary ( ) of the ground truth, respectively. This metric is calculated as follows:

where  is the Euclidean distance.

> Page 9 294 ~ 298 (of revised manuscript) Add ‘ASSD’ in the ‘Metrics for Evaluation’ section

  1. P9, L288: “…the fine-tuned model yields almost significantly (p < 0.05) better performance…”. Where are the p-values listed? And “almost significantly” is too vague. Please state the p-value explicitly. P>0.05 is not significant.

Response 10: In the Figure 4, we show the statistical significance with p-values (p<0.05) according to brain ROI region as star symbol (*) with various colors. (purple: U-Net / orange: U-Net++ / blue: FastSurfer / yellow: QuickNAT).  We performed the two-sided t-test between volume according to brain ROI regions of SAU-Net fine-tuned model and the other each model. (We excepted for SAU-Net pre-trained model since it was trained using the same dataset) The U-Net, U-Net++, and QuickNAT show statistical significance in most of the total brain ROI regions, but FastSurfer is difficult to confirm statistical significance. (p>0.05 in more than half of the total brain ROI regions)

> Page 9 288 ~ 297 Modify the paragraph

> Page 10 321 ~ Page 11 338 (of revised manuscript) Add and modify the paragraph

  1. Figures 4 and 5 are too small to be readable. Consider splitting them in sub-parts.

Response 11: We agree with the reviewer’s comment. Figure 4 and 5 were redesigned in revised manuscript.

> Page 10 Redesign Figure 4

> Page 11 Redesign Figure 5

  1. P9, L295-6: “…our fine-tuned SAU-Net model outperforms FastSurfer by an average margin (1.5%, 0.131 mm) that is clearly not statistically significant”. I do not understand this sentence. How can SAU-Net be said to outperform FastSurfer, when the margin is “clearly not significant”?

Response 12: We suggest referring to Response 10

  1. P9, L307: How exactly was “Device Inter-variability” tested? Please describe in the Methods. I also do not understand why a comparison of different scanners is relevant in a manuscript comparing segmentation methods.

Response 13: We described of “Device Inter-variability” in the revised manuscript. And, the reason for “Device Inter-variability” is to verify through comparison with other methods that it is not only accurate but also consistently robust in various devices. It is very important to train the dataset from multiple MRI scanners for robust training.

> Page 11 420 ~ 423 (of revised manuscript): Add the “Statistical Analysis” and sub-title is “Device Inter-Variability”

  1. Table 4: Use “MRI scanner” instead of “Device”. Why are some numbers bold?

Response 14: We modfied the word to “MRI scanner” and the reason for using bold is to verify that our models are consistently robust rather than the other methods among a variety of devices.

  1. P11, L322: “We can observe the under-inclusion of the left and right lateral ventricle…”. This is, for me, impossible to detect in Figure 6.

Response 15: We added the visualized figure with zoom-in. (Figure 7) In Figure 7, you can not see the blue color region (Lateral Ventricle) in yellow box of U-Net, U-Net++ and FastSurfer. However, blue color region can be detected at same location of SAU-Net and QuickNAT.

> Page 19 Line 527 (of revised manuscript): Add the Figure 7

  1. P12, Evaluation of Segmentation Reliability: this experiment should be explained in detail in the Methods section, not in the Results. It is also difficult to understand exactly what was calculated.

Response 16: We added the metric (Coefficient of Variation) and details of segmentation reliability to part of “Metrics for Evaluation”.

> Page 8 293 ~ Page 9 304 (of revised manuscript) Add the details and metric of segmentation reliability

  1. Tables 5 and 6: why was FastSurfer chosen for the comparison, and not any of the other methods?

Response 17: Since FastSufer presented the best performance amongst SOTA deep learning methods, we have conducted further analysis to compare reliability of our proposed method. We attempted to quantify the inter-session and inter-center reliability using CoRR-BNU1 and ChinaSet-Multi Center data. Additionally, the performance of U-Net little outperforms FastSurfer, however, this architecture has not novelty as SOTA deep learning method and overly implemented depending on NiftyNet library.

Reviewer 2 Report

General Comments

The authors developed a novel method for the automatic segmentation of brain areas using a 3D fully convolutional network. The method is quite interesting and shows some degree of novelty.

However, its performances are comparable to the best state of the art methods (considering the results described by the authors in the text). One of the main critical issues is related to the rationale behind the development of a novel method. It is not clear why the authors developed their approach and which gaps, with respect to the reference approaches, they want to fill.

Some methodological aspects are weak in this version of the manuscript and must be improved. For example, the description of the statistical analysis methodology is completely missing. Analogously, more details are needed regarding the MR images that were used for training and testing.

Introduction

line 78. Please correct the sentence "widely developed to applied to...." There's something wrong.

The rationale is missing. Please specify why a new method was proposed for a task that was already widely invesigated by other groups.

Materials and Methods

The authors state that 9 datasets were used, however, just 8 were listed in the text and in table 1. Please correct the document.

line 130. What is "our dataset"? Is it the CABI? Please clarify. It is not clear if the dataset is composed of 48 subjects (as described in the text) or 24 (as listed in the table). Please clarify this aspect.

Please add more information about the MR images that were included in the analysis. It would be needed to insert a new table with some features like: MR sequence (T1w, T2w, etc.) and geometrical properties (resolution, slices).

Why different zer-padding parameters were used for training (16x16x16) and testing (24x24x24)?

line 143. "MRI is a non-scaled imaging technique"...this sentence is non precise. MRI includes a lot of quantitative sequences (es. DTI, Dixon, T1/T2 relaxometry) that provide quantitative results and if the methods are standardized they can be compared among different centers.  Of course, in the case of T1W and T2w images, the story is different. Please reformulate the sentence.

Please describe better the Elastic deformation methods.

Why different learning rates were used for SAU-Net Training?

A paragraph describing the statistical analysis is missing. Please include it in the text.

How the other reference segmentation technique were implemeted? Please describe at least some essential information or refer to the specific papers.

Please, include how the assessement metrics were calculated.

Results

Table 3. Please include the statistical significance of each technique with respect to SAU-NET fine-tuned.

Figures 4 and 5 are difficult to read. Please consider splitting each of them in 2 different images to improve readibility.

Did you perform a statistical analysis also for device inter-variability. Please include this in table 4.

What is the last row of table 4?

Figure 6 doesn't show any significant difference among methods? What should the figure must say to the reader?

How the speed was calculated? Do you consider a single volume or an entire dataset? Please include all the details that are needed to better understand this aspect.

Experiments 2 and 3. Why SAU-Net was compared with just FustSurfer? What about the other methods?

Conclusions

The conclusions must be at least soften. It is not so evident that the method is overperforming the other state of the art methods.

Author Response

Response to Reviewer 2 Comments

We are grateful to the reviewer for their time and constructive comments on our manuscript. We have implemented their comments and suggestions and wish to submit a revised version of the manuscript for further consideration in the journal.

Manuscript ID brainsci-992710 entitled "Split-Attention U-Net: A fully convolutional network for robust multi-label segmentation from brain MRI"

Comments and Suggestions for Authors

In this work, the authors present a convolutional neural network for segmentation of MR images of the brain. The network was trained using six databases of annotated images. The accuracy of the segmentations was evaluated and compared to other, state-of-the-art methods. Also, the computational speed was evaluated and compared to the state-of-the-art methods. The authors conclude that the proposed method achieves superior performance compared to existing methods, while being faster than many of state-of-the-art methods.

General comments

The authors developed a novel method for the automatic segmentation of brain areas using a 3D fully convolutional network. The method is quite interesting and shows some degree of novelty.

However, its performances are comparable to the best state of the art methods (considering the results described by the authors in the text). One of the main critical issues is related to the rationale behind the development of a novel method. It is not clear why the authors developed their approach and which gaps, with respect to the reference approaches, they want to fill.

Some methodological aspects are weak in this version of the manuscript and must be improved. For example, the description of the statistical analysis methodology is completely missing. Analogously, more details are needed regarding the MR images that were used for training and testing.

Response of General Comments:

Thank you for providing us with the opportunity to revise our paper again. We hope that you find our responses satisfactory and that the manuscript is now acceptable for publication. Details appear below.

1) We extensively modified the “Introduction” with our method’s rationale

2) We splitted the “Results” to “Metrics for Evaluation”, “Statistical Analysis” and “Experimental Settings” of the “Materials and Methods”

Specific points

* of revised manuscript: revised manuscript with all TRACK

Introduction

line 78. Please correct the sentence "widely developed to applied to...." There's something wrong.

Response 1: We modified the sentence according to the reviewer’s comments as follows.

>Line 78 (of revised manuscript): Modify the sentence

The rationale is missing. Please specify why a new method was proposed for a task that was already widely invesigated by other groups.

Response 2: We added and modified the paragraph about our method’s rationale according to the reviewer’s comments as follows.

> Line 161 ~ 181 (of revised manuscript): Add the paragraph

Materials and Methods

The authors state that 9 datasets were used, however, just 8 were listed in the text and in table 1. Please correct the document.

Response 3: We corrected the wrong numbers of datasets and Table 1 was modified.

> Line 120 ~ 130 (of revised manuscript): Modify the paragraph

> Line 131 (of revised manuscript): Modify the Table 1

line 130. What is "our dataset"? Is it the CABI? Please clarify. It is not clear if the dataset is composed of 48 subjects (as described in the text) or 24 (as listed in the table). Please clarify this aspect.

Response 4: A “our dataset” was corrected to “mixed dataset” with the detail explain in the parentheses. And the paragraph of of dataset was modified, extensively.

> Line 120 ~ 146 (of revised manuscript): Modify the paragraph corresponding to dataset

> Line 131, 146 (of revised manuscript): Modify the Table 1, 2

Please add more information about the MR images that were included in the analysis. It would be needed to insert a new table with some features like: MR sequence (T1w, T2w, etc.) and geometrical properties (resolution, slices).

Response 5: We suggest referring to Response 3, 4

Why different zer-padding parameters were used for training (16x16x16) and testing (24x24x24)?

Response 6: When performing the aggregation with the inference patch, we set the patch overlap size to 12x12x12. This is an experimental defined parameter which is achieved the best Dice overlap score, and the inference was performed by setting twice the size of the patch overlap in consideration of the maximum 6 connectivity around each 3D patch, which is a situation where the patch overlaps.

line 143. "MRI is a non-scaled imaging technique"...this sentence is non precise. MRI includes a lot of quantitative sequences (es. DTI, Dixon, T1/T2 relaxometry) that provide quantitative results and if the methods are standardized they can be compared among different centers.  Of course, in the case of T1W and T2w images, the story is different. Please reformulate the sentence.

Response 7: We removed the “non-scaled imaging technique...” since this sentence is non precise and is controversial. So, we modified this sentence.

> Line 154 ~ 155 (of revised manuscript): Modify the sentence

Please describe better the Elastic deformation methods.

Response 8: A probabilistic spin was given to the basic affine transform. Since the Affine transform is linear, the image is uniformly deformed as linearity. On the other hand, elastic deformation deforms the image in different directions for each pixel. It is also common in medical imaging data since it is the data that observed living things.

Why different learning rates were used for SAU-Net Training?

Response 9: The reason for reducing the learning rate in fine-tuning is to more strongly converge some patches that have not yet converged because the training has already been sufficiently trained with pre-training

A paragraph describing the statistical analysis is missing. Please include it in the text.

Response 10: We described the statistical analysis.

> Line 316 ~ 339 (of revised manuscript): Add the paragraph

How the other reference segmentation technique were implemeted? Please describe at least some essential information or refer to the specific papers.

Response 11: The U-Net and U-Net++'s implementation has already been described (2.5 Experimental settings), and FastSurfer and QuickNAT directly applied the model file and source code published by the author to our mixed dataset to extract the segmentation results.

QuickNAt: https://github.com/ai-med/quickNAT_pytorch

FastSurfer: https://github.com/Deep-MI/FastSurfer

Please, include how the assessement metrics were calculated.

Response 10: We described the assessment metrics.

> Line 341 ~ 364 (of revised manuscrip): Add the paragraph

Results

Table 3. Please include the statistical significance of each technique with respect to SAU-NET fine-tuned.

Response 11:  We added the statistical significance in Table 3.

Figures 4 and 5 are difficult to read. Please consider splitting each of them in 2 different images to improve readibility.

Response 12: We modified the Figure 4 and 5 to enhance the readability.

> Line 407, 420 (of revised manuscript): Modify the Figure 4 and 5

Did you perform a statistical analysis also for device inter-variability. Please include this in table 4.

Response 13: Since it is two scans per device, we determined that it was not meant to get the p-value.

What is the last row of table 4?

Response 14: We inserted the word “Total” that means the overall Dice overlap mean and standard deviation of all cases.

> Line 481 (of revised manuscript): Insert the word

Figure 6 doesn't show any significant difference among methods? What should the figure must say to the reader?

Response 15: We added the Figure 7 with enlarged view to enhance the readability. We indicate two important subcortical structures, the left (blue) lateral ventricles, with yellow boxes. We can observe the under-estimation of the left lateral ventricle in the results obtained by U-Net, U-Net++, and FastSurfer. We also often observe specific classified regions where aliasing appears in the results of U-Net, QuickNAT, and FastSurfer but not in the results of U-Net++ and SAU-Net (indicated by red dashed circles) in Figure 7.

> Line 453 (of revised manuscript): Add the Figure 7

How the speed was calculated? Do you consider a single volume or an entire dataset? Please include all the details that are needed to better understand this aspect.

Response 16: The speed was calculated when a single volume were used. The runtime of the final result was measured by performing all preprocessing, inference, and aggregation. The time of the atlas-based method is excerpted from a reference paper.

> Line 344 ~ 346 (of revised manuscript): Add the paragraph

Experiments 2 and 3. Why SAU-Net was compared with just FustSurfer? What about the other methods?

Reponse 17: Since FastSufer presented the best performance amongst SOTA deep learning methods, we have conducted further analysis to compare reliability of our proposed method. We attempted to quantify the inter-session and inter-center reliability using CoRR-BNU1 and ChinaSet-Multi Center data. Additionally, the performance of U-Net little outperforms FastSurfer, however, this architecture has not novelty as SOTA deep learning method and overly implemented depending on NiftyNet library.

Conclusions

The conclusions must be at least soften. It is not so evident that the method is overperforming the other state of the art methods.

Response 18:  We modified the “Conclusions” which was more soften.

> Line 589 ~ 599 (of revised manuscript): Modify the Conclusions

Round 2

Reviewer 2 Report

The paper was extensively revised by the authors following the comments of the first-round review.